# Efficient and Stable Grouped RL Training for Large Language Models

## Abstract

Group-based reinforcement learning algorithms such as Group Reward Policy Optimization (GRPO) have proven effective for fine-tuning large language models (LLMs) with human feedback. However, generating and storing multiple completions per prompt incurs substantial memory overhead, especially as the sample group size increases, limiting scalability under constrained hardware. We propose Infinite Sampling, a framework that enables efficient and stable GRPO training by decoupling group size from GPU memory usage. It consists of: (1) *micro sampling groups* that decomposes large groups into memory-feasible rounds; (2) *continuous sampling* that interleaves generation across groups to improve utilization; and (3) a *length-aware scheduler* combining token-conditioned sequence length prediction with a two-stage plan: global grouping via fixed-point approximation scheme (FPTAS) and runtime refill via shortest-job-first (SJF). Experiments show that our *micro sampling groups* reduce peak memory usage by over 50% compared to full-group decoding (e.g., from 21.55 GB to 10.64 GB on Qwen3-1.7B). Building on this, Infinite Sampling improves throughput by over 25% compared to the sequential micro sampling group method, reducing decoding steps while maintaining full-length completions and memory usage. Our hybrid scheduling ensures efficient and stable GRPO training with larger groups under realistic GPU memory constraints.

## 1 Introduction

Large language models (LLMs), like GPT (Radford et al., 2019; Brown et al., 2020), Llama (Touvron et al., 2023a;b; Grattafiori et al., 2024), or DeepSeek (DeepSeek-AI et al., 2025), fine-tuned with reinforcement learning from human feedback (RLHF) have achieved state-of-the-art performance in aligning AI outputs with human intent. A common and effective approach in this setting is *Group Reward Policy Optimization (GRPO)* (Shao et al., 2024), which generates multiple completions per prompt and uses their aggregated rewards to stabilize policy updates. Earlier RLHF pipelines, such as InstructGPT (Ouyang et al., 2022a), relied on Proximal Policy Optimization (PPO) (Schulman et al., 2017), but methods like GRPO simplify optimization by using reward baselines across sampled groups and eliminating the need for critic networks.

However, scaling the group size $G$ — the number of sampled completions per prompt in GRPO training workflow — is memory-intensive, especially during autoregressive decoding where each output requires maintaining a separate KV cache. This memory bottleneck often prevents practitioners from utilizing large group sizes, thereby limiting the effectiveness of GRPO.

To overcome this limitation, we propose **Infinite Sampling**, a novel framework for enabling large-group GRPO training under constrained memory. Our approach builds upon the idea of *sequential micro sampling (SMS) groups*, where the full group size $G$ is decomposed into smaller subgroups decoded sequentially. While this technique allows KV cache reuse across micro groups and reduces memory usage, it introduces idle periods between consecutive decoding stages, harming throughput.

To mitigate these inefficiencies, we introduce *continuous sampling*, a new decoding paradigm inspired by continuous batching in LLM inference systems such as vLLM (Kwon et al., 2023) and SGLang (Zheng et al., 2024). Unlike continuous batching, which dynamically combines multiple unrelated user requests (Zheng et al., 2025), continuous sampling stitches together *outputs from the*

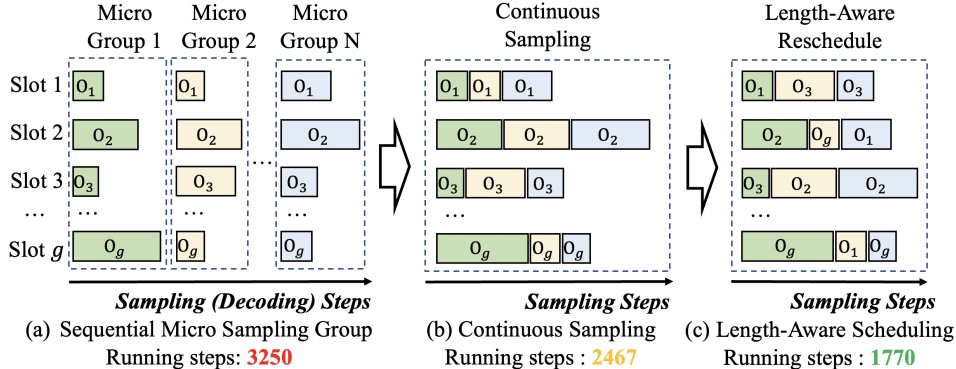

Figure 1: Illustration of the evolution from the Sequential Micro Sampling Group baseline to the final **Infinite Sampling**. (a) **Sequential Micro Sampling Group:** Samples are divided into fixed-size micro groups, which are decoded sequentially to reuse shared KV cache but underutilize available compute. (b) **Continuous Sampling:** Token-level interleaving across micro groups reduces idle time and improves throughput. (c) **Length-Aware Scheduling:** A predictive scheduler estimates prefix-conditioned sequence lengths and reorders decoding, further optimizing throughput. Here, $O$ indicates a rollout completion, and the sampling steps denote the total number of decoding iterations across all active slots (i.e., each step corresponds to generating one token per slot). Fewer sampling steps indicate higher throughput under the same total group size.

*same prompt* across micro groups using a shared prefill KV cache. This enables *interleaved decoding* of multiple micro groups, significantly reducing idle time and improving GPU utilization.

However, sequentially interleaving micro groups can trigger memory spikes when multiple long sequences are decoded concurrently. To mitigate this, we propose a two-stage length-aware scheduling strategy. First, we estimate sequence lengths by sampling a short prefix of each completion and conditioning prediction on these early tokens. This token-conditioned estimation—akin to speculative decoding (Qiu et al., 2024)—yields significantly reliable length signals. Next, we combine this with a hybrid scheduling algorithm: a global grouping phase based on a *fixed-point approximation scheme* (FPTAS), followed by a slot-level *shortest-job-first (SJF)* refill policy during decoding. This design dynamically prioritizes short sequences, achieving a favorable balance between throughput and memory stability.

Our contributions are summarized as follows:

1. We propose **Infinite Sampling**, a general framework for efficient GRPO training under memory constraints. It decouples sample group size from GPU memory usage with two techniques: (1) *micro sampling groups*, which decompose large groups into memory-feasible decoding rounds, and (2) *continuous sampling*, which interleaves generation across micro groups using a shared prompt cache.

2. We design a **length-aware decoding scheduler** to address runtime inefficiencies introduced by micro sampling. It combines prefix-based length prediction with a two-stage scheduling strategy: a global fixed-point approximation (FPTAS) for balanced group formation, and a slot-level shortest-job-first (SJF) refill policy for dynamic slot reuse.

3. We implement and evaluate Infinite Sampling on GRPO training with state-of-the-art LLMs (Qwen3 1.7B/8B). Our *Infinite Sampling* reduce peak memory usage by over **50%** compared to full-group decoding (e.g., from 21.55 GB to 10.64 GB on Qwen3-1.7B with group size 32), enabling training with large groups under tight memory budgets. On top of this, *sequential micro sampling* improves decoding throughput by avoiding full-group parallelism, while *continuous sampling* further reduces decoding steps by mitigating idle time. Finally, our full Infinite Sampling framework—with slot-level scheduling—achieves up to **45%** reduction in decoding steps (e.g., 3250 to 1770 on GSM8K), while preserving full-length completions to ensure stable GRPO training.

## 2 PRELIMINARY

**Group Relative Policy Optimization (GRPO).** GRPO (Shao et al., 2024) extends PPO (Schulman et al., 2017) by removing the value model and estimating advantages from a group of sampled com-

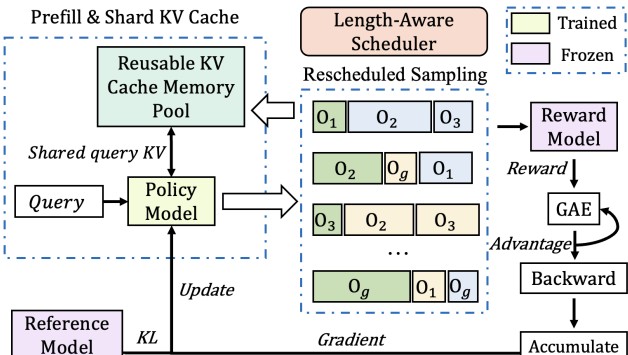

Figure 2: Overview of **Infinite Sampling**. Given a query, all completions share a prefilled KV cache. **(1)** Micro Sampling Groups (colored) partition large groups into memory-feasible rounds. **(2)** Continuous Sampling interleaves token-level decoding to maximize slot utilization. **(3)** Length-Aware Scheduling predicts prefix-conditioned lengths and orchestrates decoding via global grouping (FPTAS) and slot-level SJF refill. Generated sequences are scored and updated through standard GRPO training.

pletions. Larger groups improve estimation quality but proportionally increase memory consumption due to separate KV caches per sequence. This tension motivates our *Infinite Sampling* framework, which decouples $G$ from memory usage via micro-batching, interleaved decoding, and scheduling strategies.

**LLM Decoding via KV Cache.** Autoregressive LLMs accelerate decoding by caching key/value (KV) tensors, but this cache grows linearly with sequence length, depth, and the number of active sequences. In group-based training, where $G$ completions are sampled per prompt, maintaining $G$ KV caches quickly dominates GPU memory, forming a critical scalability bottleneck.

**Remark.** For clarity, we only highlight the essential concepts of KV caching and GRPO here. Full derivations of the GRPO objective, detailed advantage estimation, and extended discussion of memory implications are deferred to Appendix C.

## 3 METHOD

We propose **Infinite Sampling**, a decoding framework that enables large-group GRPO training under tight memory budgets. As illustrated in Figure 2, our framework consists of three components: (1) *Micro Sampling Groups*, which decompose large groups into memory-feasible decoding rounds; (2) *Continuous Sampling*, which streams token-level generation across samples to fully utilize decoding slots; and (3) a *Length-Aware Scheduling* module that predicts token-conditioned sequence lengths and orchestrates both global group planning and reactive runtime scheduling. This design enables high-throughput sampling while maintaining tight control over KV memory footprint and sample-level parallelism.

### 3.1 MICRO SAMPLING GROUP: MEMORY-EFFICIENT SAMPLING

Generating a large number of samples $G$ per prompt, as required in GRPO, incurs a prohibitive memory cost. This is because autoregressive decoding allocates a separate KV cache buffer for each sampled sequence, and the memory footprint scales linearly with $G$. Directly decoding all $G$ sequences in parallel quickly exceeds the available GPU memory, particularly for long sequences.

To address this challenge, we propose to decompose the full sample group into $N$ smaller *micro sampling groups*, each of size $g = G/N$. Instead of allocating memory for all $G$ samples at once, we decode only one micro group at a time and reuse a shared memory region for the KV cache across groups. This approach caps the peak decoding memory at the cost of a single micro group, enabling us to support larger effective group sizes without exceeding hardware limits.

**KV Cache Pooling.** We maintain a fixed-size memory pool capable of storing the KV cache for up to $g$ active sequences. This pool is initialized before sampling begins and reused across all micro groups. During the decoding of a micro group, its KV cache is dynamically written into this pool.

Once decoding finishes for the current group, its cache is cleared and the memory is reassigned back to the pool. Importantly, we retain the prefill KV cache for the prompt itself, which is shared by all groups. This allows us to avoid recomputing the prompt context and only manage memory for the completion portion of each group.

Figure 1(a) illustrates this process. At each stage, the memory allocated for one micro group is overwritten by the next, enabling bounded-memory decoding. This scheme offers a simple yet effective trade-off: although we introduce some sequential processing (one group at a time), we gain the ability to scale to arbitrarily large group sizes within fixed memory.

## 3.2 CONTINUOUS SAMPLING: INTERLEAVED GENERATION STRATEGIES

While micro sampling enables memory-efficient decoding, it introduces a fundamental throughput bottleneck due to its sequential execution pattern. Specifically, in our micro sampling design, each group is decoded one after another to stay within the memory budget. Although samples within a micro group are decoded in parallel, *inter-group barriers* introduce idle GPU slots when short sequences finish early and must wait for longer ones to complete before proceeding to the next group. This leads to underutilization of available compute, especially under high sample-length variance.

To address this, we introduce **continuous sampling**, a decoding paradigm that interleaves token generation across samples to maintain high utilization of decoding slots while preserving the memory efficiency of micro sampling. This strategy is enabled by the fact that all completions in GRPO originate from the same prompt and hence share the same prefill context. Figure 1(b) illustrates the token-level interleaving enabled by continuous sampling.

**Two Modes of Continuous Sampling.** Our Continuous sampling can be implemented in two distinct modes, each with its own trade-offs:

(1) *Fixed-Slot Continuous Sampling (Figure 3(a))*. In this mode, the total group of size $G$ is still divided into $N$ micro groups, each of fixed size $g = G/N$. Over the $N$ micro-group rounds, each decoding slot outputs one sequence per round, thus $N$ sequences in total. As soon as any sequence in the group completes, a new sequence is launched in its place, forming the next micro group in the following decoding step. This design maintains a uniform group size at each stage, facilitating micro-batched reward computation and consistent slot usage. Importantly, micro groups may overlap in time—there is no need to wait for the entire group to finish before starting the next. This design enables structured yet flexible decoding aligned with group-based training pipelines.

(2) *Dynamic-Slot Continuous Sampling (Figure 3(b))*. In contrast, this mode does not enforce uniform micro group sizes. Instead, decoding proceeds in a fully streaming fashion: as soon as a sequence completes, its slot is immediately reassigned to a new sample, regardless of how many are currently active. The only constraint is that a total of $G$ sequences must be generated in the end. This strategy maximizes slot throughput but sacrifices structural regularity, making it less compatible with micro-batched updates. Moreover, dynamic-slot scheduling may introduce length bias, where shorter sequences are favored due to faster turnover.

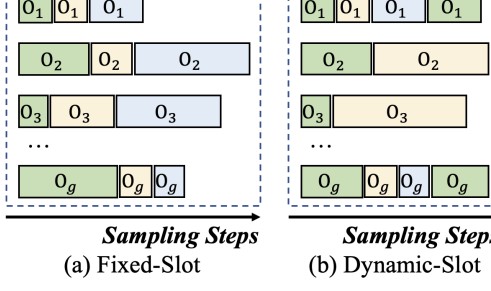

*Sampling Steps*  
(a) Fixed-Slot  

*Sampling Steps*  
(b) Dynamic-Slot

Figure 3: **Two Modes of Continuous Sampling.** (a) In Fixed-Slot mode, exactly $N$ sequences will be generated for each row, maintaining consistent slot utilization but potentially suffering from idle slots due to length variance. (b) In Dynamic-Slot mode, slots are immediately reused as soon as a sequence finishes, maximizing utilization but sacrificing group structure.

**Discussion and Implications.** Our system supports both fixed-slot and dynamic-slot continuous sampling. The fixed-slot variant provides consistent micro-batched structure and fairness, but may suffer from idle compute when short sequences must wait for longer ones in the same group. The dynamic-slot variant improves responsiveness and throughput by fully interleaving decoding, yet introduces potential bias favoring shorter sequences. To alleviate the efficiency bottlenecks in fixed-slot sampling, we introduce a *length-aware scheduling strategy* in Section 3.3. By estimating

sequence lengths ahead of time and balancing group assignments accordingly, this scheduler smooths per-slot decoding time and maximizes utilization under the fixed-slot setting.

**Shared Prompt KV and Sample-Level Cache Lifecycle.** As in Section 3.1, all completions share the same prompt KV cache computed during the prefill phase. Each active sample maintains a separate KV buffer for its completion tokens, which is recycled upon completion to support new samples within bounded memory.

---

**Algorithm 1** Two-Stage Length-Aware Scheduling for Fixed-Slot Sampling

---

**Require:** Estimated lengths $\{\hat{l}_1, \ldots, \hat{l}_G\}$, number of slots $g$
**Ensure:** Decoding execution plan under fixed-slot sampling with $g$ active slots

 1: **Phase 1: Static Micro Group Assignment**
 2: Apply FPTAS-Based Group Assignment (Algorithm 2): $\mathtt{mask}[i] \leftarrow (n, j)$

 3: **Phase 2: Runtime Decoding with SJF Refill**
 4: Initialize active slots $\mathcal{S} \leftarrow$ first $g$ samples from $\mathtt{mask}$
 5: **while** not all samples are completed **do**
 6:     **for** each active slot $s \in \mathcal{S}$ in parallel **do**
 7:         Decode one token for current sample
 8:         **if** sample in slot $s$ finishes **then**
 9:             Mark sample as completed and release $s$
10:             Apply Slot-Level SJF Refill (Algorithm 3) to select next sample for $s$
11:         **end if**
12:     **end for**
13: **end while**

---

**Comparison with Continuous Batching.** At first glance, our continuous sampling method (Figure 5b) may appear similar to continuous batching techniques (Figure 6) developed for LLM inference systems, such as vLLM (Kwon et al., 2023) and SGLang (Zheng et al., 2024). However, the two differ in both purpose and implementation. Continuous batching targets *multi-user inference* workloads where each request originates from a distinct prompt. These systems merge unrelated prompts and decode them in shared compute steps, relying on asynchronous request arrival. In contrast, our setting is tailored for *training-time sampling* in GRPO, where all completions are derived from a single prompt. This enables us to compute the prompt KV cache once and reuse it across all samples—an optimization that is not applicable in inference batching. Furthermore, we maintain full control over the sampling loop and cache lifecycle, allowing for token-aware, per-sample scheduling and cache recycling. These characteristics enable more aggressive memory reuse and scheduling strategies, beyond what continuous batching systems support.

### 3.3 LENGTH-AWARE SCHEDULING: FROM STATIC GROUP PRE-PLANNING TO ADAPTIVE RUNTIME SCHEDULING

While continuous sampling interleaves decoding across micro groups to improve throughput, it introduces a new challenge: simultaneous decoding of multiple long sequences can still lead to memory spikes due to cumulative KV cache usage. Compared to the micro group strategy described in Section 3.1, the fixed-slot continuous sampling mode in Section 3.2 improves GPU utilization by filling all slots continuously. However, it remains limited by the "longest sequence" effect—i.e., shorter sequences must wait for the longest one in the end, constraining overall throughput. To address this, we propose a **two-stage scheduling strategy** that combines *offline prefix-based global planning* with *online memory-aware dynamic scheduling*, explicitly designed to mitigate this bottleneck under the fixed-slot constraint.

**Prefix Sampling and Length Estimation.** To estimate completion lengths before full decoding, we first sample a short prefix (e.g., the first $k$ tokens) for each output and predict its final length using a token-conditioned estimator. In contrast to inference-time length prediction—where prompts vary across samples—our setting generates multiple completions from the same prompt. Therefore, prompt-only predictors fail to distinguish among different completions. To address this, we concatenate the prefix to the original prompt to form a pseudo-prompt, preserving the individuality of

each sample. This pseudo-prompt is then passed to the predictor. This design provides lightweight yet informative length signals for scheduling. We defer detailed implementation of the predictor to Section 4.

---

**Algorithm 2** FPTAS-Based Micro Group Assignment

---

**Require:** Estimated lengths $\{\hat{l}_1, \ldots, \hat{l}_G\}$, number of groups $N$, tolerance $\epsilon$
**Ensure:** Mapping $\texttt{mask}[i] \leftarrow (n, j)$ assigning sample $i$ to group $n$ at position $j$
1: $S \leftarrow \sum_{i=1}^{G} \hat{l}_i, \quad K \leftarrow \epsilon \cdot S/N$
2: **for** $i = 1$ to $G$ **do**
3: $\quad \tilde{l}_i \leftarrow \lceil \hat{l}_i/K \rceil$
4: **end for**
5: $\tilde{C} \leftarrow \lceil \sum_{i=1}^{G} \tilde{l}_i/N \rceil$
6: Initialize $L_n \leftarrow 0$ (total load of group $n$), $\mathcal{G}_n \leftarrow \emptyset$ for $n = 1$ to $N$
7: **for** each sample $i$ sorted by descending $\tilde{l}_i$ **do**
8: $\quad$ **for** $n = 1$ to $N$ **do**
9: $\quad\quad$ **if** $L_n + \tilde{l}_i \leq \tilde{C}$ **then**
10: $\quad\quad\quad$ Assign $i$ to $\mathcal{G}_n$ with position $j \leftarrow |\mathcal{G}_n|$
11: $\quad\quad\quad$ Set $\texttt{mask}[i] \leftarrow (n, j)$
12: $\quad\quad\quad$ Update $L_n \leftarrow L_n + \tilde{l}_i$
13: $\quad\quad\quad$ **break**
14: $\quad\quad$ **end if**
15: $\quad$ **end for**
16: **end for**

---

**Stage 1: Static Group Pre-Planning via FPTAS.** To construct memory-efficient decoding plans prior to generation, we formulate micro group assignment as a classical multiprocessor scheduling problem: given a set of tasks (samples) with estimated execution costs $\{\hat{l}_1, \ldots, \hat{l}_G\}$, we aim to partition them into $N$ bins (micro groups) such that the total cost in each bin remains below a predefined memory threshold, and the overall makespan—i.e., the maximum memory load across groups—is minimized. To this end, we adapt a fixed-point approximation scheme (FPTAS) (Algorithm 2) for bin packing with additive error tolerance $\epsilon$. This produces a near-optimal grouping plan that balances memory usage across groups while respecting system constraints. Unlike heuristic clustering (e.g., greedy or round-robin), FPTAS ensures formal guarantees on group balance and minimizes the risk of decoding bottlenecks due to misaligned group assignments. This stage produces a static execution plan that guides the initial sampling order and provides a strong baseline for memory-bounded decoding.

**Stage 2: Runtime Adjustment via Shortest-Job-First (SJF).** While the FPTAS provides a globally optimized plan, static scheduling cannot account for runtime variability such as early termination or mispredicted sequence lengths. To handle these dynamics, we introduce a slot-level online adjustment mechanism based on a shortest-job-first (SJF) (Algorithm 3) policy. During continuous sampling, decoding proceeds in an interleaved fashion, and the number of active slots is bounded by available memory. Whenever a sequence completes and releases its KV cache, we immediately dispatch a new sample into the freed slot. The SJF policy ranks all pending samples by their predicted lengths $\hat{l}_i$, prioritizing shorter samples to maximize slot turnover and reduce memory contention. Importantly, this adjustment is non-blocking and globally aware: samples from future micro groups may be promoted early if they fit the current memory profile. This transforms the decoding process into a fluid, slot-level stream scheduler that adapts to actual completion signals, amortizes memory spikes, and exploits idle compute across groups.

Together, the global planning of FPTAS and the fine-grained responsiveness of SJF yield a decoding pipeline that is both stable and agile—achieving near-optimal memory distribution while remaining robust to length prediction noise and sampling variance.

**Benefits.** This hybrid strategy balances structured planning with runtime flexibility. Prefix-conditioned predictions allow approximate scheduling without requiring deterministic sampling trajectories, while SJF adjustment ensures robustness to length prediction errors. The combination allows us to scale to large group sizes with stable memory usage, fast slot refill, and high throughput.

A schematic overview of our two-stage scheduling results is shown in Figure 1(c). We summarize the complete procedure in Algorithm 1.

---

**Algorithm 3** Shortest-Job-First (SJF) Refill

---

**Require:** Finished slot $s$, estimated lengths $\{\hat{l}_i\}$, finished set $\mathcal{F}$, mask `mask`
**Ensure:** Assign next sample to slot $s$
  1: $\mathcal{C} \leftarrow \{i \mid i \notin \mathcal{F} \wedge i \notin \text{mask}\}$
  2: **if** $\mathcal{C} = \emptyset$ **then**
  3:      **return** no refill
  4: **end if**
  5: Select $j \leftarrow \arg\min_{i \in \mathcal{C}} \hat{l}_i$
  6: Assign $j$ to slot $s$ at next position $\text{pos}[s]$
  7: Update $\text{mask}[j] \leftarrow (s, \text{pos}[s])$; increment $\text{pos}[s]$

---

## 4   SAMPLING LENGTH PREDICTION

### 4.1   MOTIVATION AND DIFFERENCE FROM PRIOR WORK

Existing work on sequence length prediction (e.g., Qiu et al. (2024); Choi et al. (2025); Guldogan et al. (2024)) primarily focuses on inference settings, where the output length is predicted from the input prompt. These methods assume a one-to-one mapping between prompt and output, which does not hold in our training scenario. In Group Reward Policy Optimization (GRPO), all samples within a group share the same prompt, and the variation arises only from the stochastic sampling process. As a result, prompt-only predictors fail to distinguish among different completions. To address this, we introduce a token-conditioned estimation strategy that accounts for each sample's early generation trajectory.

### 4.2   TOKEN-CONDITIONED PREDICTION VIA PSEUDO-PROMPTS

To obtain per-sample predictions, we first decode a short prefix of $k$ tokens from the current policy for each sample. These tokens are then concatenated to the original prompt, forming a pseudo-prompt:

$$\hat{l}_i = f_{\text{reg}}(\text{BERT}([x; O_{1:k}]))$$

where $x$ is the original prompt, and $O_{1:k}$ denotes the first $k$ tokens decoded from the policy for sample $i$. The pseudo-prompt is passed to a pretrained BERT encoder, and the [CLS] token embedding is projected by a regression head to estimate the expected length $\hat{l}_i$.

We fine-tune a length prediction model using the LMSYS-Chat dataset (Zheng et al., 2023), and adopt a standard BERT encoder architecture (Devlin et al., 2019) for token-conditioned pseudo-prompts regression. During decoding, the BERT model is used in frozen inference mode to estimate lengths with negligible runtime overhead. This setup yields accurate and efficient scheduling signals in practice.

**Prefix Reuse Optimization.** Since the first $k$ tokens have already been decoded during length estimation, we reuse them during actual decoding to avoid redundant computation. For each sample, decoding resumes from token $k+1$, with its KV cache initialized from the prefix. This design eliminates wasted computation and ensures consistency between estimation and execution.

## 5   EXPERIMENTS

### 5.1   EXPERIMENTAL SETUP

We evaluate Infinite Sampling on three tasks: **GSM8K** (Cobbe et al., 2021), **MATH** (Lightman et al., 2023), and **KK** (Xie et al., 2025). All experiments are run on NVIDIA A100-80GB GPUs using **Qwen3** (Yang et al., 2025) models of size 1.7B and 8B.

Unless otherwise noted, we set prompt batch size $B = 1$, group Size $G = 32$, Micro Group Size $g = 4$, maximum generation length to 1024 tokens, and generation temperature is 0.8. All reported metrics are averaged over all prompts from all datasets. *Memory* refers to the peak GPU memory usage during decoding (lower is better), and *Sampling Steps* denotes the total number of decoding steps required to complete all sequences (lower indicates faster decoding). Each sampling step corresponds to one token-generation round across all $g$ active decoding slots—that is, $g$ tokens

are generated in parallel in each step. Therefore, the total number of sampling steps reflects how efficiently the decoding slots are utilized. Fewer steps indicate better throughput under fixed memory and compute budgets. Due to space limitations, we leave more experiments in Appendix Section E.

## 5.2 MAIN RESULTS

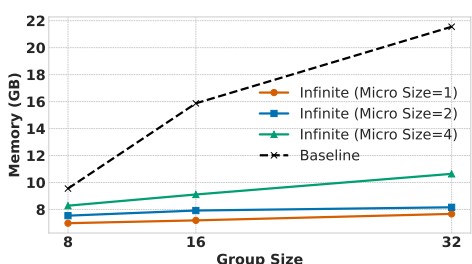
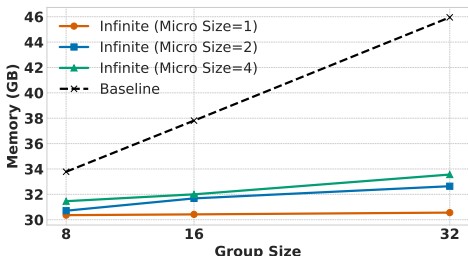

(a) Qwen3-1.7B: Infinite Sampling consistently reduces memory usage across micro group sizes.

(b) Qwen3-8B: Even for larger models, Infinite Sampling maintains significant memory savings.

Figure 4: Peak memory usage comparison under different group sizes and micro group configurations for Infinite Sampling. The baseline refers to directly decoding all $G$ sequences in parallel. Infinite Sampling achieves substantial memory savings across models.

Our primary goal is to reduce memory overhead in group-based reinforcement learning by decomposing large sampling groups into memory-efficient micro groups. We compare the peak GPU memory usage of Infinite Sampling under different micro group sizes ($g = 1, 2, 4$) against the baseline vLLM that decodes all $G = 8, 16, 32$ samples in parallel.

Figure 4 reports results on Qwen3-1.7B and Qwen3-8B. Across all configurations, Infinite Sampling consistently reduces memory consumption, with smaller micro groups yielding greater savings. For example, on Qwen3-1.7B with $G = 32$, the baseline requires 21.55 GB of memory, while Infinite Sampling with $g = 1$ reduces this to 10.64 GB, a 51% reduction. Importantly, baseline memory usage grows nearly linearly with the group size $G$, reflecting the additive cost of KV caches per sample. In contrast, Infinite Sampling maintains almost constant memory usage across different $G$, as decoding is bounded by the fixed micro group size $g$. These results confirm that micro sampling enables scaling to large group sizes without exceeding hardware memory limits.

## 5.3 ABLATION STUDY

We compare four decoding strategies introduced in Section 3: **Sequential Micro Group** (Section 3.1), **Fixed-slot** (Section 3.2), **Dynamic-slot** (Section 3.2), and **Infinite-sampling** (Section 3.3). Here, "**Infinite-sampling \***" represents a theoretical oracle performance or a reference lower bound, where all completions, with known length, are scheduled post hoc after generation, allowing the shortest possible sampling steps.

Table 1 reports sampling steps and average sequence length across three datasets and two model sizes. We make the following key observations: **Decoding Latency**: Across all tasks, Sequential Micro Group consistently results in the highest number of decoding steps. For instance, on the MATH dataset with Qwen3-1.7B, it requires 7005 steps, while *Infinite-sampling* reduces this to only 5203 steps (a 25.7% improvement). On KK, the reduction is even more pronounced—7037 (*Sequential*) to 5200 (*Infinite-sampling*), a 26.1% gain. **Sequence Length Preservation**: *Infinite-sampling* and *Fixed-slot* both preserve the average sequence length, matching that of the *Sequential Micro Group*. For example, on the GSM8K dataset with Qwen3-8B, all three strategies produce an average of 186 tokens, ensuring sample quality is not sacrificed. **Dynamic-Slot Tradeoff**: Although Dynamic-slot achieves the fewest decoding steps (e.g., 360 steps on GSM8K with Qwen3-8B), it substantially shortens the generated sequences (e.g., only 21 tokens on average). As discussed in Section 3.2, this is due to length bias introduced by streaming generation and greedy slot reuse, which may negatively impact training stability and reward quality. **Offline Scheduling Optimality**: On KK with Qwen3-8B, *Infinite-sampling\** achieves 2586 steps compared to 2599 for the actual Infinite-sampling implementation—demonstrating that the practical scheduling is near-optimal (only 0.5% higher). **Stability vs. Efficiency**: *Infinite-sampling* provides a balanced trade-off, maintaining long outputs

Table 1: **Ablation Study.** Sampling steps and average sequence length of different proposed components. This serves as a lower-bound reference rather than a practical decoding strategy. All subsequent tables include this reference for comparison.

| Dataset | Method | Qwen3-1.7B | | Qwen3-8B | |
|---|---|---|---|---|---|
| | | Sampling Steps | Avg Length | Sampling Steps | Avg Length |
| GSM8K | SMS Group | 3250 | 186 | 3250 | 186 |
| | Fixed-slot | 2467 (x0.75) | 186 (x1.00) | 2467 (x0.75) | 186 (x1.00) |
| | Dynamic-slot | 748 (x0.23) | 72 (x0.39) | 360 (x0.11) | 21 (x0.11) |
| | Infinite-sampling * | 1739 (x0.53) | 186 (x1.00) | 1739 (x0.53) | 186 (x1.00) |
| | Infinite-sampling | 1770 (x0.54) | 186 (x1.00) | 1769 (x0.54) | 186 (x1.00) |
| MATH | SMS Group | 7005 | 602 | 5051 | 395 |
| | Fixed-slot | 6098 (x0.87) | 602 (x1.00) | 4190 (x0.83) | 395 (x1.00) |
| | Dynamic-slot | 1395 (x0.20) | 143 (x0.23) | 1626 (x0.32) | 172 (x0.43) |
| | Infinite-sampling * | 5126 (x0.73) | 602 (x1.00) | 3492 (x0.69) | 395 (x1.00) |
| | Infinite-sampling | 5203 (x0.74) | 602 (x1.00) | 3533 (x0.70) | 395 (x1.00) |
| KK | SMS Group | 7037 | 591 | 4682 | 282 |
| | Fixed-slot | 6129 (x0.87) | 591 (x1.00) | 3550 (x0.75) | 282 (x1.00) |
| | Dynamic-slot | 1104 (x0.16) | 102 (x0.17) | 696 (x0.15) | 38 (x0.13) |
| | Infinite-sampling * | 5089 (x0.72) | 591 (x1.00) | 2586 (x0.55) | 282 (x1.00) |
| | Infinite-sampling | 5200 (x0.74) | 591 (x1.00) | 2599 (x0.56) | 282 (x1.00) |

while reducing latency. In contrast, *Dynamic-slot* aggressively optimizes for throughput at the cost of sequence quality, which may be unsuitable for reward-based learning. These results highlight the trade-offs between decoding efficiency and sequence quality. While *Dynamic-slot* is fastest, *Infinite-sampling* offers a better balance between speed and GRPO effectiveness.

# 6 RELATED WORK

RLHF has become the standard paradigm for aligning LLMs, with early pipelines such as Instruct-GPT relying on PPO (Ouyang et al., 2022b; Schulman et al., 2017). Recent methods including DPO (Rafailov et al., 2024) and GRPO (Shao et al., 2024) simplify training by removing critic networks and normalizing rewards across sampled groups. While effective, scaling group size $G$ is often limited by KV cache memory overhead. System-level work has focused on efficient RLHF infrastructures (Sheng et al., 2025; Zhong et al., 2025; Fu et al., 2025) and inference-oriented batching with KV cache reuse (Kwon et al., 2023; Zheng et al., 2025). Orthogonal efforts such as ZeRO (Rajbhandari et al., 2020), ZeRO-Infinity (Rajbhandari et al., 2021), and FlashAttention (Dao et al., 2022; Dao, 2023) target memory efficiency during training. However, none directly address decoding-time memory in group-based RLHF. Our work is the first to combine micro sampling, continuous decoding, and length-aware scheduling to enable large-group GRPO training under strict memory budgets. For a complete survey and positioning, please refer to the full related work in Appendix F.

# 7 CONCLUSION

We present **Infinite Sampling**, a framework for enabling large-group GRPO training under constrained GPU memory. By decomposing full sample groups into memory-feasible micro groups, interleaving decoding across groups, and applying token-conditioned length-aware scheduling, our method substantially reduces peak memory usage while preserving sample quality. While our approach introduces partial serialization and does not match the theoretical latency of fully parallel decoding, we design continuous sampling and hybrid scheduling (FPTAS + SJF) to mitigate the overhead. This achieves a favorable trade-off between memory efficiency and decoding throughput, enabling stable and scalable GRPO training with large group sizes under realistic hardware constraints. While Infinite Sampling achieves strong memory efficiency and decoding performance, our current implementation assumes all completions originate from a single prompt. Extending our framework to support multi-prompt batches or integrating with more advanced KV cache compression techniques is an interesting direction for future work. We hope this work offers a practical foundation for memory-efficient group-based RLHF optimization at scale.

## ETHICS STATEMENT

Our work focuses on improving the efficiency of reinforcement learning algorithms for large language models. We do not introduce new data or modify reward functions, but rather optimize the sampling and scheduling process. As such, the ethical risks are similar to those already present in RLHF training, including potential biases or harmful outputs learned from underlying datasets. We emphasize that our contributions are system-level optimizations and do not directly alter model behavior. Future deployment should follow established best practices for safe data curation, alignment, and responsible use of LLMs.

## REPRODUCIBILITY STATEMENT

We aim to ensure reproducibility by (1) providing detailed algorithmic descriptions of Infinite Sampling (Sections 3–4 and Appendix) and (2) reporting full experimental settings including datasets, model sizes, hyperparameters, and evaluation metrics (Section 5 and Appendix).

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

APPENDIX

## A  THE USE OF LARGE LANGUAGE MODELS (LLMS)

Large Language Models (LLMs) were employed for grammar correction and text refinement to improve the clarity and readability of this paper.

## B  MORE FIGURES

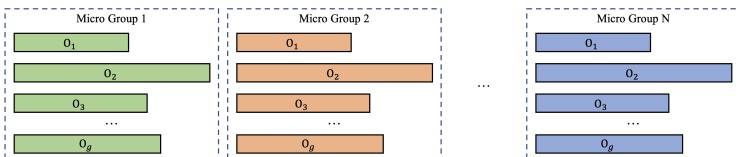

(a) **Sequential Micro Sampling Group.** Each micro group is decoded after the previous one finishes. This allows KV memory reuse but results in under-utilization due to sequential execution.

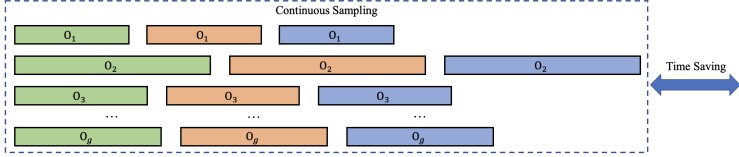

(b) **Continuous Sampling.** We interleave micro group completions using a shared prompt KV cache, avoiding idle time and improving memory efficiency.

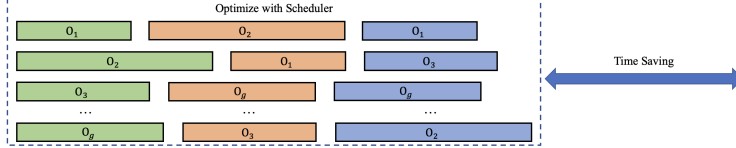

(c) **Length-Aware Scheduling.** Samples are first length-predicted using token-conditioned prefixes, then dynamically scheduled to smooth memory usage and improve slot turnover.

Figure 5: Illustration of the three key components in **Infinite Sampling**. (a) **Sequential Micro Sampling Group:** Samples are divided into fixed-size micro groups, which are decoded sequentially to reuse shared KV cache but underutilize available compute. (b) **Continuous Sampling:** Token-level interleaving across micro groups reduces idle time and improves throughput. (c) **Length-Aware Scheduling:** A predictive scheduler estimates prefix-conditioned sequence lengths and reorders sequences decoding, further optimizing memory usage and latency.

## C  FULL PRELIMINARY

### C.1  LLM DECODING VIA KV CACHE

Autoregressive language models decode text token-by-token by attending to previously generated tokens. To avoid recomputing attention over the entire prefix at each step, modern LLMs cache key and value (KV) tensors from past layers during decoding. This mechanism, known as the KV cache (Pope et al., 2022; Shi et al., 2024), significantly accelerates inference by reusing previously computed attention states.

During decoding, each new token requires a forward pass through all transformer layers, where the current token attends to both cached and current representations. As a result, the KV cache grows linearly with the sequence length and model depth. In typical LLM implementations, each completion maintains a dedicated KV buffer across all decoder layers and heads.

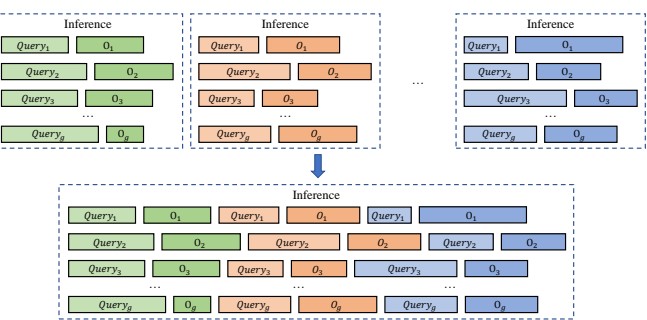

Figure 6: Continuous Batching in inference merges unrelated prompts and completions to eliminate bubbles. Not directly usable in GRPO.

While KV caching improves efficiency, it introduces substantial memory overhead when decoding multiple sequences concurrently. In GRPO-style training, where $G$ completions are generated per prompt, the total memory footprint scales linearly with $G$, sequence length, and model size. This makes KV cache the primary memory bottleneck when training with large sample groups.

## C.2    GROUP RELATIVE POLICY OPTIMIZATION (GRPO) FOR LLMS

Group Reward Policy Optimization (GRPO) (Shao et al., 2024) is an enhanced variant of the widely used policy gradient reinforcement learning algorithm, Proxy Policy Gradient (PPO) (Schulman et al., 2017), which is specially designed to improve memory and computation efficiency. PPO relies on a value model to predict the state value of a generated completion for advantage calculation (Schulman et al., 2018), where the value model is often initialized with a pretrained LLM of comparable size to the policy model and jointly trained in using supervised learning. Instead, GRPO eliminates the need for a value model by estimating advantages using a Monte Carlo method. It calculates advantages based on the rewards of a group of randomly sampled outputs, significantly reducing memory and computational overhead while ensuring effective policy optimization.

Specifically, GRPO generates a group of outputs for a given prompt $x$ by randomly sampling from the policy $\pi_\theta$:

$$\{O_1, O_2, \ldots, O_G\} \sim \pi_\theta(\cdot \mid x), \tag{1}$$

where, $G$ is a hyperparameter that determines the group size, i.e., the number of outputs sampled for each prompt $x$. The advantages are estimated using the normalized rewards within the group as:

$$A_i = \frac{r_i - \bar{\mathbf{r}}}{\sigma(\mathbf{r})}, i \in \{1, 2, \ldots, G\}, \tag{2}$$

where $\mathbf{r} = \{r_1, r_2, \ldots, r_G\}$ represents the rewards corresponding to the group of sampled outputs. These rewards can be derived either from a rule-based reward function or a learned reward model. $\bar{\mathbf{r}}$ and $\sigma(\mathbf{r})$ denote the mean and standard deviation of the rewards in the group, respectively. Based on the estimated advantages, the policy model in GRPO is optimized similarly to PPO, using the following optimization objective

$$\mathcal{J}_{\text{GRPO}}(\theta) = \frac{1}{G}\sum_{i=1}^{G}\frac{1}{|O_i|}\sum_{t=1}^{|O_i|}\Big\{ \min\Big[\lambda_t(\theta)A_{i,t},\ \text{clip}\big(\lambda_t(\theta),\ 1-\varepsilon,\ 1+\varepsilon\big)A_{i,t}\Big]$$
$$- \beta\,\mathbb{D}_{KL}\big[\pi_\theta \,\|\, \pi_{\text{ref}}\big]\Big\}, \tag{3}$$

where $\lambda_t(\theta) = \frac{\pi_\theta(o_{i,t}|x,O_{i,<t})}{\pi_{\theta_{\text{old}}}(O_{i,t}|x,O_{i,<t})}$ is the ratio of predicted probability of token $o_{i,t}$ under the current policy model $\theta$ to that under the old policy model $\theta_{\text{old}}$, which was used to sample the output group. $A_{i,t} = A_i$ for $t \in \{1, 2, \ldots, |O_i|\}$, where $|O_i|$ is the number of tokens in output $O_i$. $\varepsilon$ is a hyperparameter controlling the clipping range for conservative updates, and $\beta$ is a hyperparameter controlling the weight of the KL-divergence constraint with respect to a reference policy model $\pi_{\text{ref}}$.

For further details on the optimization objective, we recommend referring to the PPO and GRPO paper.

The group size $G$ is a critical hyperparameter that significantly impacts the estimation of advantages, thereby influencing the optimization stability and the performance of the resulting policy model. According to standard Monte Carlo principles (Metropolis & Ulam, 1949; Hastings, 1970), increasing the group size typically improves the accuracy of advantage estimation by incorporating more samples. However, scaling up the group size $G$ introduces substantial overhead. In particular, larger $G$ leads to higher memory consumption during autoregressive decoding, as each sample requires a separate KV cache. These constraints place significant pressure on GPU memory, often making large-group training impractical in real-world settings. To overcome this limitation, we propose *Infinite Sampling*, a framework that decouples the group size $G$ from memory usage by introducing micro-batching, token-level interleaving, and dynamic scheduling strategies.

# D    REWARD COMPUTATION AND MICRO-BATCHED POLICY UPDATE

While our main focus is on efficient decoding, the subsequent stages—reward computation, advantage normalization, and policy gradient updates—must also be adapted to the micro sampling setup. Both the reward aggregation and the backward pass are executed in micro batches aligned with the decoding schedule. This ensures consistent memory usage throughout the full GRPO pipeline (Figure 2).

However, the decoding stage remains the dominant cost, especially when generating long sequences. As a result, our method section emphasizes decoding-side optimizations. For completeness, we describe the reward and training stages in detail.

After decoding, each sampled sequence $O_i$ is scored with a scalar reward $r_i$, computed as a combination of a reward model score and a KL penalty relative to a reference policy $\pi_{\theta_{\text{ref}}}$:

$$r_i = \text{RM}(O_i, x) - \beta \cdot \log \frac{\pi_\theta(O_i \mid x)}{\pi_{\theta_{\text{ref}}}(O_i \mid x)}$$

As mentioned before, we decompose the full sample group of size $G$ into $N$ micro groups:

$$G = N \cdot g, \quad \text{where } g \text{ is the micro group size.}$$

We compute rewards $\{r_i\}_{i=1}^G$ and corresponding advantages in a streaming fashion, processing each micro group $\mathcal{G}^{(n)} = \{O_{i_n}, \ldots, O_{i_n+g-1}\}$ independently.

**Group-Normalized Advantage per Micro Batch.** Although micro groups are processed sequentially, the GRPO formulation requires group-wide normalization. Therefore, we cache per-sample rewards from each micro group and compute the full-group mean reward after all $N$ micro batches, as done in normal GRPO:

$$A_i = r_i - \frac{1}{G} \sum_{j=1}^{G} r_j.$$

This ensures that gradient updates still reflect the entire sample group distribution.

**Micro-Batched Backpropagation.** After computing $A_i$ for all $i \in [1, G]$, we launch backpropagation in micro batches. For each micro group $\mathcal{G}^{(n)}$, we compute the token-level GRPO loss:

$$\mathcal{J}_{\text{GRPO}}^{(n)}(\theta) = \frac{1}{g} \sum_{O_i \in \mathcal{G}^{(n)}} \frac{1}{|O_i|} \sum_{t=1}^{|O_i|} \Big\{ \min\Big[\lambda_t(\theta) A_{i,t}, \ \text{clip}\big(\lambda_t(\theta), \ 1-\varepsilon, \ 1+\varepsilon\big) A_{i,t}\Big] \tag{4}$$
$$- \beta \cdot \mathbb{D}_{\text{KL}}\big[\pi_\theta \,\|\, \pi_{\theta_{\text{ref}}}\big] \Big\},$$

where $\lambda_t(\theta) = \frac{\pi_\theta(o_{i,t}|x,o_{i,<t})}{\pi_{\theta_{\text{old}}}(o_{i,t}|x,o_{i,<t})}$ is the token-level importance weight.

Backpropagation is performed incrementally on each $\mathcal{J}_{\text{GRPO}}^{(n)}(\theta)$, enabling memory-efficient training with arbitrarily large group size $G$.

The total loss can be calculated across all micro groups:

$$\mathcal{J}_{\text{GRPO}}(\theta) = \frac{1}{N} \sum_{n=1}^{N} \mathcal{J}_{\text{GRPO}}^{(n)}(\theta).$$

**Memory Efficiency.** This micro-batched pipeline ensures that neither decoding, reward computation, nor backpropagation requires instantiating all $G$ completions in memory simultaneously. It complements Infinite Sampling's decoding design and enables end-to-end GRPO training with large group sizes under tight memory budgets.

# E MORE EXPERIMENTS

## E.1 SCHEDULER CHOICE

Table 2: Sampling steps of different scheduling methods.

| Dataset | Schedule Method | Qwen3-1.7B Sampling Steps | Qwen3-8B Sampling Steps |
|---------|-----------------|---------------------------|-------------------------|
| GSM8K | Fixed-slot | 2467 | 2467 |
| | Infinite-sampling * | 1739 (x0.70) | 1739 (x0.70) |
| | FPTAS only | 2100 (x0.85) | 2100 (x0.85) |
| | SJF only | 1775 (x0.72) | 1774 (x0.72) |
| | Infinite-sampling | 1770 (x0.72) | 1769 (x0.72) |
| MATH | Fixed-slot | 6098 | 4190 |
| | Infinite-sampling * | 5126 (x0.84) | 3492 (x0.83) |
| | FPTAS only | 6021 (x0.98) | 4154 (x0.99) |
| | SJF only | 5380 (x0.88) | 3624 (x0.86) |
| | Infinite-sampling | 5203 (x0.85) | 3533 (x0.84) |
| KK | Fixed-slot | 6129 | 3550 |
| | Infinite-sampling * | 5089 (x0.83) | 2586 (x0.72) |
| | FPTAS only | 6014 (x0.98) | 3531 (x0.99) |
| | SJF only | 5219 (x0.85) | 2695 (x0.76) |
| | Infinite-sampling | 5200 (x0.84) | 2599 (x0.73) |

We study the impact of different scheduling components introduced in Section 3.3. Table 2 reports the number of sampling steps under the Fixed-slot setting, using four variants: **Fixed-slot** (no scheduling), **FPTAS only** (only uses static pre-grouping), **SJF only** (only uses slot-level dynamic refill), and **Infinite** (FPTAS + SJF).

We make the following key observations:

**Slot-Level Scheduling Dominates**: SJF-only consistently yields the greatest reduction in decoding steps among all individual strategies. For instance, on GSM8K with Qwen3-1.7B, SJF reduces the steps from 2467 (Fixed-slot) to 1775, a 28.1% improvement. Similar trends are observed across other datasets, confirming that online dynamic refill is the key factor in improving efficiency.

**FPTAS Provides Secondary Gains**: The FPTAS-only strategy offers moderate gains over Fixed-slot (e.g., 2100 vs. 2467 on GSM8K), but consistently underperforms SJF. This aligns with its design: FPTAS statically balances load before decoding, which improves average-case slot utilization but cannot adapt to runtime variation. These results suggest that FPTAS is useful but insufficient on its own, because, as discussed in Section 3.3, it cannot adapt to runtime variability such as early termination or prediction noise. In contrast, SJF dynamically corrects these deviations during decoding, quickly filling idle slots with short sequences. These results highlight that runtime correction is crucial for achieving high decoding throughput and robust memory usage.

**Infinite Matches the Lower Bound**: The full Infinite scheduling strategy (FPTAS + SJF) consistently performs within 1% of the oracle Infinite-sampling *, which assumes access to all completions before

scheduling. For example, on KK with Qwen3-8B, Infinite achieves 2599 steps, while the oracle baseline reaches 2586. This demonstrates that combining global planning with runtime reactivity nearly closes the gap to optimality.

### E.2 EFFECT OF PREFIX LENGTH $k$ ON PREDICTION ACCURACY

To investigate the impact of prefix length on the accuracy of length prediction (Section 4), we evaluate our BERT-based length predictor under different values of $k \in \{1, 2, 4, 8, 16, 32\}$, where $k$ denotes the number of initial tokens sampled from the policy model before estimation.

Figure 7 reports the prediction error across two model scales: Qwen3-1.7B and Qwen3-8B. We observe that increasing $k$ consistently improves prediction accuracy up to a point, with diminishing returns beyond $k = 16$. Notably, the Qwen3-8B model achieves better accuracy than Qwen3-1.7B at every $k$, likely due to its more consistent output distributions.

Based on these results, we use $k = 16$ as the default prefix length for length prediction in all Infinite Sampling experiments reported in the main paper.

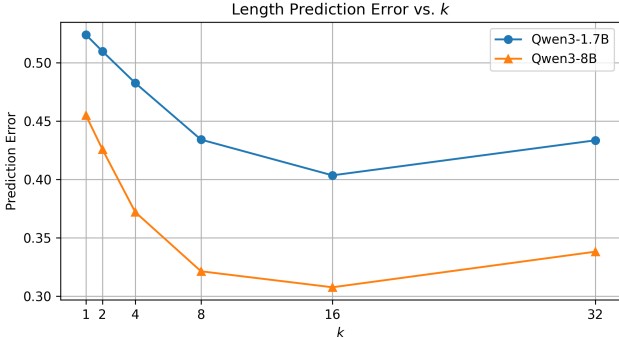

Figure 7: Length prediction error vs. prefix length $k$. Qwen3-8B consistently achieves lower error, and accuracy improves as more tokens are included for prediction, up to $k = 16$.

## F   FULL RELATED WORK

**RLHF and Grouped Sampling Optimization.** Reinforcement Learning from Human Feedback (RLHF) has become the de facto approach for aligning large language models (LLMs) with human preferences. Early pipelines such as InstructGPT (Ouyang et al., 2022b) rely on Proximal Policy Optimization (PPO) (Schulman et al., 2017) to fine-tune policies using scalar rewards from a learned reward model. However, PPO introduces complexity due to the use of critic networks and unstable advantage estimation. To mitigate these issues, recent methods such as Direct Preference Optimization (DPO) (Rafailov et al., 2024) and Group Reward Policy Optimization (GRPO) propose critic-free, group-normalized objectives. GRPO in particular improves training stability by aggregating scores over multiple sampled completions per prompt. Our work builds directly on group-based reinforcement learning algorithms, such as GRPO, and tackles the underexplored question of how to support large group sizes $G$ under strict memory constraints—a challenge not addressed in prior GRPO literature.

**Efficient RLHF Frameworks.** Recent RLHF frameworks focus on building general-purpose and efficient infrastructure for running RLHF pipelines. HybridFlow (Sheng et al., 2025) introduces a hybrid programming model that decouples intra-node model computation (using multi-controller execution) and inter-node data orchestration (using a centralized controller), achieving efficient model placement and resharding across training and generation stages. StreamRL (Zhong et al., 2025) and AReaL (Fu et al., 2025) further streamline RLHF actor rollout by improving token-level parallelism and offloading efficiency. In contrast, Infinite-sampling does not propose a general-purpose RLHF framework. Instead, it focuses on decoding efficiency by introducing micro sampling groups and continuous scheduling to maximize GPU utilization and memory throughput during actor rollout. While existing frameworks aim to optimize end-to-end RLHF pipelines through system-level

modularity and flexibility, our work complements them by proposing fine-grained decoding strategies that can be integrated into these infrastructures. Our techniques are lightweight and inference-centric, designed to decrease sampling memory usage without requiring changes to reward modeling, training loops, or inter-model protocols.

**Batching and KV Cache Optimization.** A growing body of work focuses on improving memory efficiency and throughput in LLM inference through KV cache management and dynamic batching. Notably, vLLM (Kwon et al., 2023) introduces PagedAttention and continuous batching to reuse KV cache and maximize hardware utilization. BatchLLM (Zheng et al., 2025) further enhances cache reuse through prefix-aware batching and reorder-based scheduling. However, these systems are inference-centric and assume independent user requests. Our work adapts these ideas to the training-time setting of RLHF, where all completions originate from the same prompt and must be grouped for reward aggregation. We propose a continuous sampling mechanism tailored for such homogeneous input settings, while also incorporating length-aware scheduling for memory control—something not explored in existing batched decoding systems.

**Memory-Efficient LLM Training.** Orthogonal to inference efficiency, many efforts have improved memory utilization during training, such as ZeRO (Rajbhandari et al., 2020), ZeRO-Infinity (Rajbhandari et al., 2021), and FlashAttention (Dao et al., 2022; Dao, 2023). These methods target optimizer state sharding, parameter offloading, and fused GPU kernels to reduce memory and bandwidth overheads. Our approach is complementary: rather than optimizing backpropagation or attention itself, we focus on *reducing decoding-time memory* during RLHF training by restructuring how and when sample completions are generated. To our knowledge, we are the first to combine micro sampling, continuous sampling, and slot-level scheduling to support large-group GRPO training under fixed memory budgets.

**Summary.** In sum, while prior work has studied reward aggregation in RLHF, sequence length prediction for inference, and memory-efficient model execution, none have addressed the unique combination of challenges posed by large-group GRPO training. Our method, **Infinite Sampling**, is the first to jointly address group-level decoding, token-interleaved generation, and adaptive sample scheduling, offering a unified framework for scalable and stable RLHF under memory constraints.

