# OpenReview forum: "Efficient and Stable Grouped RL Training for Large Language Models"
_ICLR.cc/2026/Conference — ICLR 2026 Conference Withdrawn Submission_

### Official Review · Reviewer_KiRB · 2025-10-30

**Soundness:** 2
**Presentation:** 2
**Contribution:** 2
**Rating:** 2
**Confidence:** 4

**Summary:**

This paper addresses the memory bottleneck in Group Reward Policy Optimization (GRPO) for fine-tuning large language models, where generating multiple completions per prompt with separate KV caches creates prohibitive memory consumption. The authors propose Infinite Sampling, a framework comprising three components: (1) partitioning sampling batches into smaller sequential decoding units that enable KV cache reuse across subgroups; (2) enabling concurrent token generation through micro-group interleaving with shared prompt encoding to eliminate idle periods; and (3) employing early-token-based length forecasting coupled with dual-phase scheduling—FPTAS for balanced initial grouping and SJF for dynamic slot allocation—to prevent memory spikes. Experiments demonstrate over 50% peak memory reduction, 25% throughput improvement, and up to 45% fewer decoding steps while maintaining stable GRPO training with full-length completions.

**Strengths:**

- The paper is well-structured with a logical progression from problem identification to solution design.
- The paper makes valuable contributions by approaching GRPO optimization through a systems lens rather than purely algorithmic modifications.
- The experimental results demonstrate substantial practical gains though there are some limitations.

**Weaknesses:**

- A major concern is that the proposed methods may not meaningfully improve upon state-of-the-art system implementations already deployed in practice. Modern RLHF systems commonly replicate prompts $G$ times without prefix sharing mechanisms to mitigate GPU out-of-memory issues. Given this context, it remains unclear whether Sequential Micro Sampling Groups and Continuous Sampling provide substantial advantages over existing optimized implementations.

- The length-aware scheduling strategy critically depends on a pre-trained length predictor, which introduces significant practical limitations. For many real-world applications—particularly training math reasoning models with long-tail response length distributions—obtaining reliable length predictions is challenging or infeasible. Response lengths are inherently model-dependent and evolve throughout training as the policy updates, making pre-trained predictors quickly obsolete.

**Questions:**

- The experimental setup appears limited in scale, with batch size restricted to $B=1$ and maximum generation length capped at 1024 tokens. However, production RLHF systems typically operate with substantially larger configurations: batch sizes of $B=128$ or $B=512$, and maximum generation lengths extending to 16K tokens or beyond, particularly for reasoning-intensive tasks. These scale differences could fundamentally alter the performance characteristics and bottlenecks of the system. Could the authors provide evidence that the proposed methods remain effective and produce notable efficiency gains under realistic production-scale settings? Specifically, how do memory savings, throughput improvements, and scheduling overhead scale with larger batch sizes and longer sequences?

- The paper would benefit from explicit benchmarking against modern, optimized RLHF training frameworks. Have the authors compared Infinite Sampling with state-of-the-art implementations such as Verl, OpenRLHF, or similar systems that already incorporate advanced memory optimization techniques? Understanding the incremental benefits over existing production-grade systems is crucial for assessing the practical contribution of this work.


- Recent advances in RLHF have explored asynchronous training settings where generation and training are decoupled to improve resource utilization. Can the proposed techniques—particularly continuous sampling and length-aware scheduling—remain valuable or be adapted to asynchronous training configurations? Clarifying the scope of applicability would help position this work within the evolving landscape of RLHF system designs.

---

### Official Review · Reviewer_HjGr · 2025-10-31

**Soundness:** 3
**Presentation:** 3
**Contribution:** 3
**Rating:** 4
**Confidence:** 3

**Summary:**

This paper introduces Infinite Sampling, a framework to enable efficient and stable group-based reinforcement learning (GRPO) for large language models (LLMs) under limited GPU memory.
 The authors identify a key bottleneck in GRPO — the linear growth of KV-cache memory with group size G, since each sampled completion maintains its own cache during autoregressive decoding.
The proposed framework has three main components:
- Micro Sampling Groups (MSG): Decomposes large groups into smaller micro-groups that reuse a shared prompt KV cache, drastically reducing peak memory usage.
- Continuous Sampling (CS): Introduces token-level interleaving across micro-groups to maintain high GPU utilization.
- Length-Aware Scheduling (LAS): Predicts completion lengths via prefix-based estimation and applies a two-stage scheduling strategy (FPTAS + SJF) for efficient runtime balancing.

**Strengths:**

- Strong motivation and practical relevance: The memory bottleneck in GRPO is a real limitation for RLHF practitioners; this paper addresses it directly.

- System–algorithm co-design: The modular approach (MSG + CS + LAS) is both conceptually elegant and implementable.

- Novelty in decoding-side optimization: Continuous Sampling and token-conditioned scheduling are new contributions to RLHF training rather than inference.

- Comprehensive experiments: Multiple datasets, model scales, and ablations demonstrate consistent improvements and near-oracle scheduling performance.

**Weaknesses:**

Limited end-to-end measurement: The paper reports decoding steps and memory but does not show overall wall-clock training speed or total throughput (including reward computation and backprop).


Simplified length predictor: The BERT-based length regression approach seems heuristic; more justification or ablation on its generalization would strengthen the claim.

**Questions:**

- Could the prefix-based length predictor be integrated directly into the policy network instead of using a separate BERT model?


- Does token-level interleaving affect reward normalization or introduce bias in advantage estimation?


- How robust is the FPTAS + SJF scheduling under inaccurate length predictions?


- Could Infinite Sampling scale to multi-prompt batches where prompts differ but share model parameters?

---

### Official Review · Reviewer_6ejM · 2025-11-03

**Soundness:** 3
**Presentation:** 3
**Contribution:** 2
**Rating:** 6
**Confidence:** 3

**Summary:**

This paper proposes Infinite Sampling, a memory-efficient decoding framework for enabling large-group GRPO (Group Reward Policy Optimization) training under constrained GPU memory.

**Strengths:**

The problem is real and important. The writing is mostly clear.  This paper primarily focuses on system optimizations instead of algorithmic innovation, yet such improvements are by no means trivial. The execution of the paper seems solid, and I appreciate the authors for their efforts in improving the implementation of GRPO.

**Weaknesses:**

I did not notice any obvious flaws. However, I am not an expert on system optimizations, so it is possible that I missed important weaknesses.

**Questions:**

1. I am bit confused about the length-aware scheduler in Section 3.3.  The description seems long and tedious. I have one question here: With this scheduler, is the proposed method still a lossless and exact alternative to the default implementation of GRPO? Does the method guarantee to output the same model as the default implementation GRPO after an identical training run?

2. Correct me if I misunderstood: The paper primarily focused on the reduction of memory usage and decoding steps. However, the wall-clock running time comparison is missing. How does Infinite Sampling improve in terms of end-to-end training time (including reward computation and updates)?

---

### Official Review · Reviewer_QbAw · 2025-11-04

**Soundness:** 3
**Presentation:** 2
**Contribution:** 2
**Rating:** 4
**Confidence:** 5

**Summary:**

This paper introduces Infinite Sampling, a framework designed to address the prohibitive memory overhead of KV caching in GRPO for large language models (LLMs). The framework decouples group size from memory consumption by decomposing large batches into sequential Micro Sampling Groups and employs Continuous Sampling to interleave token generation, thereby maximizing GPU utilization. This process is further optimized by a hybrid Length-Aware Scheduler, which combines offline FPTAS-based planning with online SJF-based refilling to manage workloads efficiently. Experimental results indicate the method reduces peak memory usage by over 50% and improves decoding throughput by up to 45%, enabling  large-scale GRPO training on resource-constrained hardware.

**Strengths:**

1. The paper proposes a critical and highly practical bottleneck in modern LLM alignment research. The KV cache memory consumption during group-based RL methods like GRPO is a significant barrier that often prevents practitioners with limited hardware from exploring larger, more stable training configurations.

2. The highlight of the innovation is the two-stage, length-aware scheduler, which combines the near-optimal global planning of an offline approximation scheme  with the real-time flexibility of an online policy. This hybrid design creates a system that is both robust in its planning and highly adaptive to runtime dynamics.

**Weaknesses:**

1. The most significant flaw of this paper is its exclusive focus on system-level metrics (memory, throughput) while completely neglecting the impact on the reinforcement learning process itself. The paper's title claims to enable "Efficient and Stable GRPO Training," yet it provides no evidence to support the "stable" claim or demonstrate that the resulting model is effective.

2. Several of the paper's central claims are not adequately supported by the provided experiments, and the experimental setup relies on unverified assumptions.

3. The paper's contribution appears to be more of an application of existing ideas to a specific context, rather than a fundamentally new technique, and its applicability is narrow.

**Questions:**

1. Could you quantify the memory savings and performance advantage gained from sharing and pre-calculating the prompt's KV cache (Section 3.1)? Given that prompt lengths are often much shorter than completion lengths, how significant is this optimization in practice?

2. How is the sequential processing time for micro-groups calculated? More importantly, the paper is motivated by the need for "arbitrarily large group sizes", yet the experiments are limited to G=32. This seems insufficient to justify the "Infinite". Could you provide experimental evidence demonstrating how final model accuracy scales as G increases to larger values (e.g., 64, 128, 256)? This would validate the premise that larger group sizes are indeed necessary and beneficial for RL training.

3. The "Continuous Sampling" method appears functionally identical to the continuous batching used in inference engines like vLLM (L242-252), just applied within a GRPO group. Could this not be achieved through engineering optimizations within a framework like vLLM? If so, what is the core conceptual novelty of your method beyond this specific application?

4. Does the FPTAS scheduling, combined with SJF, introduce a bias that favors shorter samples and suppresses longer ones? Have you analyzed the sampling distribution of sequence lengths under this mechanism to ensure that longer completions are not unfairly deprioritized during training?

5. What dataset was used for the evaluation in Figure 7? How well does the predictor generalize to out-of-distribution (OOD) data or more complex problems (e.g., AIME), where length can be highly variable? What is the justification for using a BERT-based model to approximate the generation behavior of a much larger LLM?

6. Conditioning the prediction on the first k tokens freezes a portion of the generation, which could reduce sample diversity. How is this trade-off managed, especially when rollouts from the same prompt can naturally have vastly different lengths?

7. Why are there no accuracy comparisons in the paper? A direct comparison of final task accuracy between a baseline GRPO implementation and one using Infinite Sampling is crucial for evaluating whether the system optimizations negatively impact model quality.

8. Is the maximum generation length of 1024 tokens sufficient for complex reasoning tasks where solutions can be much longer? This choice might limit the generalizability of your findings.

---

### Note · Authors · 2026-01-23

I have read and agree with the venue's withdrawal policy on behalf of myself and my co-authors.